# A Highly Effective Bacteriophage-1252 to Control Multiple Serovars of *Salmonella enterica*

**DOI:** 10.3390/foods12040797

**Published:** 2023-02-13

**Authors:** Chuan-Wei Tung, Zabdiel Alvarado-Martínez, Zajeba Tabashsum, Arpita Aditya, Debabrata Biswas

**Affiliations:** 1Department of Animal and Avian Sciences, University of Maryland, College Park, MD 20742, USA; 2Biological Sciences Program-Molecular and Cellular Biology, University of Maryland, College Park, MD 20742, USA

**Keywords:** *Salmonella enterica*, bacteriophage, whole-genome sequencing, bioinformation, biocontrol

## Abstract

*Salmonella enterica* (*S. enterica*) is the most common foodborne pathogen worldwide, leading to massive economic loss and a significant burden on the healthcare system. The primary source of *S. enterica* remains contaminated or undercooked poultry products. Considering the number of foodborne illnesses with multiple antibiotic resistant *S. enterica*, new controlling approaches are necessary. Bacteriophage (phage) therapies have emerged as a promising alternative to controlling bacterial pathogens. However, the limitation on the lysis ability of most phages is their species-specificity to the bacterium. *S. enterica* has various serovars, and several major serovars are involved in gastrointestinal diseases in the USA. In this study, *Salmonella* bacteriophage-1252 (phage-1252) was isolated and found to have the highest lytic activity against multiple serovars of *S. enterica*, including Typhimurium, Enteritidis, Newport, Heidelberg, Kentucky, and Gallinarum. Whole-genome sequencing analysis revealed phage-1252 is a novel phage strain that belongs to the genus Duplodnaviria in the Myoviridae family, and consists of a 244,421 bp dsDNA, with a G + C content of 48.51%. Its plaque diameters are approximately 2.5 mm to 0.5 mm on the agar plate. It inhibited *Salmonella* Enteritidis growth after 6 h. The growth curve showed that the latent and rise periods were approximately 40 min and 30 min, respectively. The burst size was estimated to be 56 PFU/cell. It can stabilize and maintain original activity between 4 °C and 55 °C for 1 h. These results indicate that phage-1252 is a promising candidate for controlling multiple *S. enterica* serovars in food production.

## 1. Introduction

Diarrheal disease remains the most common illness associated with consuming contaminated foods; it is one of thetop 10 leading causes of illness, accounting for the illness of 550 million individuals and the death of 230,000 individuals each year [1,2]. In the USA, among the 25,606 cases of foodborne illnesses, 35% were caused by non-typhoidal *Salmonella enterica*. Of these illnesses, many lead to hospitalizations and deaths; when added together with loss of productivity, the cost of recovery, and the elimination of recalled products, this amounts to a massive economic burden and puts an additional burden on the healthcare system every year [3]. Previous studies have shown *S. enterica* contains more than 2610 different serovars, some of which have specific environmental niches that make them more common among specific food sources [4]. Patients that suffer from *S. enteric* infection with *S. enterica* can experience mild to moderate gastroenteritis with fever, vomiting, nausea, diarrhea, and stomach cramps [5]. However, depending on the host’s pre-existing health conditions, the infection can progress into a more severe illness, such as bacteremia, meningitis, and other focal infections [4].

The estimated prevalence ranking (from high to low) of the four major serovars of *S. enterica* that cause illnesses in the USA are Enteritidis (16.8%), Newport (10%), Typhimurium (9.8%), and Heidelberg (1.6%) [6]. Previously, Ferrari et al. (2019) also reported that the *S. enterica* serovar Enteritidis is the most common invasive non-typhoidal *Salmonella* serovar associated with human salmonellosis [7]. The most common attributable sources of salmonellosis have been found to be contaminated food, specifically chicken, egg, vegetables, and fruits [8,9]. However, animal and particularly poultry feces commonly contaminate vegetables, fruits, and other goods through water and soil pollution.

Due to intensive use of antibiotics, either as a therapeutic or growth promoter (mostly in Asian and African countries, as the USA and Europe have restricted the use of antibiotics in promoting the growth of farm animals), the rate of selection of increasingly antibiotic-resistant bacteria has accelerated [10]. These multi-drug resistant bacteria can resist a wide variety of antibiotics, which can result in the development of severe medical and therapeutic problems all over the world [11]. Current studies show there are 700,000 deaths per year due to multi-drug resistant bacteria, and it is estimated that most current antibiotics will be ineffective by 2025 [12]. The multi-drug resistant issue will cause 10 million deaths worldwide by 2050 [13], which will lead to a need to find an alternative to synthetic antibiotics.

Currently, the application of bacteriophages (also known as phages) has gained popularity worldwide as a promising option to address the control of bacterial pathogens without further contributing to the development of antimicrobial resistance [14,15]. Phages are viruses that recognize bacterial cells rather than animal or human cells. In general, phages are ubiquitous in the natural environment, where they cohabitate with bacteria, including in water sediments, soil, and even mucosa surfaces of animals and humans [16].

This study aimed to isolate and characterize novel *Salmonella* phages with lytic ability against multiple non-typhoidal *Salmonella* serovars, and evaluate their therapeutic effect against predominant causative agents of salmonellosis *in vitro*. Furthermore, the prevalence and virulence of antimicrobial resistance genes during the lytic activity of phages against *Salmonella* serovars were targeted for analysis using whole-genome sequencing technology.

## 2. Materials and Methods

### 2.1. Bacterial Strains and Culture Conditions

In this study, we used six serovars of *S. enterica*, including Enteritidis, Typhimurium, Newport, Heidelberg, Kentucky, and Gallinarum. Out of these six serovars, *S*. Typhimurium (ATCC LT2) and *S.* Enteritidis (ATCC13076) were purchased from American Type Culture Collection (ATCC, Manassas, VA, USA) and *S.* Newport, *S.* Heidelberg, *S.* Kentucky, and *S.* Gallinarum were isolated from animal farms and characterized previously in our laboratory [17]. In addition, a Shiga toxin-producing enterohemorrhagic *Escherichia coli* (*E. coli*) O157: H7 EDL933 (ATCC 700927) was also used (Table 1). All strains were previously preserved in 40% glycerol (*v*/*v*) at −80 °C and revived on Luria–Bertani (LB) agar (Becton, Dickinson and Co., Sparks, MD, USA) through incubation at 37 °C overnight.

### 2.2. Isolation and Propagation of Phages

Phages were isolated from various samples collected from animal (cattle, poultry, and turkey) farms, including drinking water, feces, and lagoons on farms in Maryland, USA, using the standard microbiological techniques and using *S.* Enteritidis as the bacterial host cell (BHC) in the study. Briefly, collected samples were centrifuged at 4500× *g* for 30 min. Then, 100 µL supernatants were added to 50 mL of BHC culture of LB broth and incubated at 37 °C for 24 h. The BHC containing LB broth was previously prepared with a starting culture of 100µL bacterial suspension (OD_600_ = 0.1) in 50 mL LB broth and incubated at 37 °C overnight. After the overnight incubation of the supernatant from the centrifuged sample and BHC containing LB broth, the tube was further centrifuged at 4500× *g* for 30 min to separate the phages suspended in the supernatant. The supernatant was then collected and filtered using a 0.22 μm sterile syringe filter (VWR, Radnor, PA, USA) to remove any remaining bacterial cells or other particles.

### 2.3. Identification and Purification of Phages

Isolation of *Salmonella*-specific phages was carried out using the biphasic agar (double-layer method) assay [18]. Briefly, a bacterial suspension for each individual serovar being studied was prepared by adjusting the suspension to an optical density (OD_600_ = 0.1, ~10^8^ CFU/mL) in LB broth. After that, 100 µL of each bacterial suspension was added to 5 mL of LB soft agar (LB with 0.5% agar) kept at 56 °C in a water bath and poured onto previously prepared LB agar plates (with 1.6% agar). After drying soft agar at room temperature, 10 µL of each phage sample isolated was spotted onto the surface of each *Salmonella* serovar plate and incubated at 37 °C for 24 h. Samples that exhibited a clear zone of inhibition on the soft agar portion of the media were considered positive samples. To purify a single phage, ten-fold serial dilutions of positive phage samples were plated with the biphasic agar layer method, and phages that produced clear plaques were selected, sub-cultured in an LB broth suspension containing specific *Salmonella* serovars and grown at 37 °C overnight. The isolated phages were sub-cultured for an additional six passages to obtain a pure phage suspension [19].

### 2.4. Host Range Analysis

The host spectrum of the newly isolated *Salmonella* phages was determined using the spot assay and the biphasic agar assay inoculated with different bacterial serovars and later expanded by cross-infecting these with phages isolated from different host bacterial serovars [20]. After the isolation and purification of the phage, 10 μL of the phage suspensions were spotted onto the top surface of bi-phasic agar plates which had a lawn of different *Salmonella* serovar and enterohemorrhagic *E. coli* (Table 1). The spot test plates were dried at room temperature and incubated overnight at 37  °C, then observed for the formation of the plaques. Plaque formation on the plates was evaluated visually and assigned a score for phage lytic ability [21].

### 2.5. Phage DNA Extraction

The genomic DNA of phage candidates, including phage-1212, phage-1223, phage-1225, phage-1252, phage-1336, phage-CF, phage-CW, and phage-2902 was purified with certain modifications described in [22]. Briefly, phages were prepared in a 450 μL phage suspension containing 1 μL DNase I (1 U/mL) and 1 μL RNase A (10 mg/mL), which was incubated at 37 °C for 1.5 h without shaking to degrade any exogenous bacterial nucleic acids. After eliminating exogenous DNA, neutralization of the DNase and RNase enzymes was achieved by adding a solution containing a mixture of 20 μL of 0.5 M EDTA, 1.25 μL of proteinase K (20 mg/mL), and 20 μL of 10% SDS, and incubating at 56 °C for further 1 h. Then, the phage DNA was isolated with a DNeasy Blood and Tissue kit (Qiagen, Germantown, MD, USA) following the manufacturer’s instructions. The quality and concentration of the phage genomic DNA were assessed through the NanoVue (Biochrom, Holliston, MA, USA) and Qubit (Invitrogen, Carlsbad, CA, USA).

### 2.6. Whole-Genome Sequencing and Bioinformatics Analysis

Whole-genome sequencing was performed on the 20 selected *Salmonella* phages. The DNA library was constructed according to the manufacturer protocol of the Illumina MiSeq system. The sequencing libraries were prepared using the Illumina Nextera XT library preparation kit. Whole-genome sequencing was analyzed by the Average Nucleotide Identity (ANI) [23] and NCBI-BLAST. Paired DNA sequencing raw reads were provided as FASTQ files in Miseq basespace, and analysis was performed using software tools of SPAdes (Galaxy Version 3.15.3 + galaxy2) and Trimmomatic (Galaxy Version 0.38.0) from GALAXY (Galaxy version 23.0.rc1, https://usegalaxy.org, accessed on 8 February 2023) [24]. The phylogenetic tree was generated by NCBI-BLAST using the complete genome sequences of phage 1252. Mauve [25] and Circular Genome Viewer (CGview) [26] were used for genome comparison at the DNA level, based on the genomic sequences available in the NCBI database. To determine the lytic ability and exclude the lysogenicity of phages, complete genomes were subjected to PHAge Search Tool Enhanced Release (PHASTER) [27] and Rapid Annotation using Subsystem Technology (RAST) [28]. To identify the virulence factors and antimicrobial resistance genes, all the genome sequencing data were screened by the platform of Virulence Factors of Pathogenic Bacteria (VFPB) [29] and the Comprehensive Antibiotic Resistance Database [30].

### 2.7. Determining the Phenotypic Characteristics of Phage-1252

The lytic activity of phage-1252 against *S. enterica* was examined in vitro [31]. Briefly, 100 μL of BHC culture (10^5^ CFU/mL) and an equal volume of phage-1252 suspension (10^5^ PFU/mL) were added to 1.8 mL LB broth in a 24-well microwell plate, to achieve multiplicity of infection (MOI) ranges from 1 to 1000. After inoculation with phage, samples were incubated at 37 °C for 24 h. A colony forming unit (CFU)/mL of BHC was measured at 0, 3, 6, 9, 12, and 24 h intervals and compared to an untreated BHC culture that was used as the control. Each assay was repeated three times.

To determine the phage growth trend, the phage burst size and latency period were determined through a one-step growth curve [32] with some modifications. The BHC was grown in LB broth at 37 °C to the OD_600_ of 0.1. Then, 100 μL of 1 × 10^7^ PFU/mL (plaque forming unit/mL) phage was added and incubated for 10 min at 37 °C to allow for phage adsorption. The absorption mixture was centrifuged at 8000 rpm at 4 °C for 10 min to collect the pellets that were later resuspended in 10 mL of LB broth (preheated to 37 °C). 100 μL resuspended pellets were then used to make ten-fold serial dilutions to a final dilution of 1 × 10^−1^, 1 × 10^−2^, and 1 × 10^−3^, and incubated for 60 min at 37 °C. During the 60 min incubation, 0.1 mL was taken from each dilution at various time points, mixed with 200 μL of BHC suspension (in LB broth), and plated using 0.5% (*w*/*v*) LB agar. Each experiment of one-step growth was repeated three times. The time frame between phage absorption and the start of the first phage burst was known as the latent period. The ultimate number of enhanced phages to the starting number of bacteria determined the burst size.

For thermal stability, the phages were evaluated their stability at different temperatures. The phage suspension was serially diluted for 105 from phage stocks for accurate count (between 30 and 300 PFU), then was incubated at 4 °C, 20 °C, 30 °C, 35 °C, 40 °C, 45 °C, 50 °C, 55 °C, and 60 °C for 1 h. A volume of 100 μL was collected from each treated phage, and their PFU was detected using the aforementioned biphasic agar. 

### 2.8. Statistical Analysis

SAS 3.8 (Enterprise Edition, SAS ONDEMAND FOR ACADEMICS) (SAS Institution Inc., Cary, NC, USA) was used to determine the statistical significance. A one-way analysis of variance (ANOVA) was applied to determine significant differences in the lytic activity control and treatment and thermal stability, based on a significance level of 0.05 (*p* < 0.05).

## 3. Results

### 3.1. Isolation of Phages and Their Diverse Host Ranges

By using *S.* Enteritidis as a bacterial host cell (BHC) and spot test, phage-positive samples showed apparent plaque and clear boundaries on the agar plate. A total of 20 *Salmonella*-specific phages were identified and isolated from collected farm samples (Appendix A). The host range spot test for these phages was further tested by using different *Salmonella* serovars and enterohaemorrhagic *E. coli* (Table 1 and Table 2) as BHC. A total of eight phage candidates were purified and isolated from the initial 20 phages, including phage-1212, phage-1223, phage-1225, phage-1252, phage-1336, phage-CF, phage-CW, and phage-2902 (Table 2). The efficiency of the lytic activity of these phages was measured against six serovars of *Salmonella*, including *S.* Typhimurium, *S.* Enteritidis, *S.* Newport, *S.* Heidelberg, *S.* Kentucky, *S.* Gallinarum, and *E. coli* O157: H7 (Table 2), using the scoring system and visual assessment of plaques on the spot test. The isolated phages were purified by six generations of purification.

### 3.2. Whole-Genome Sequences of Isolated Phages and Their Unique Patterns

The full-length genomes of eight candidate phages were analyzed using the BLAST tool from NCBI, and further studied using the GenBank database. Based on BLAST analysis, the results showed that sequences of three phage candidates (phage-1212, phage-1223, and phage-1225), isolated from pond water and feces of a cattle farm, were identifiable as Salmonella phage vB_SenM-S16 (100% identity). The remaining five phages (phages-1252, phage-1336, phage-2902, phage-CF, and phage-CW) were not found to match with known *Salmonella* phage homologies and currently known phages in GenBank. The whole-genome sequencing of phages-1252, phage-1336, phage-2902, phage-CF, and phage-CW was analyzed by the ANI calculator. The result showed phages-1252, phage-1336, and phage-2902 are one strain (100% identity), and phage-CF and phage-CW are the other strain (100% identity). The similarity of these two strains is around 97–98% (Table 3).

Phage-2902 showed the strongest lytic activity against multiple host strains compared to other isolated phage strains. Based on the BLAST, whole-genome sequencing showed several contigs (or gene segments) of phage-2902; one major contig (gene size: 243,229 bp) showed 100% identity to phage 1252, and another contig (gene size: 52, 856 bp) showed 100% identity to known phage [PEA2-3 (52,770 bp, NCBI: txid2808969). After multiple steps of isolation and purification of phage-2902 from a single plaque, the whole-genome sequencing of phage-2902 still confirmed the combination (not shown in the study). We can exclude that it was contaminated by phage PEA2-3, and can exclude any possible errors that led to it becoming a cocktail phage strain. Therefore, we precluded phage-2902 in this study because it needs further study to reveal the relationship and interaction between two phage genes and how they can infect host cells simultaneously. Based on the diversity of sequencing and lyric ability against multiple *S. enterica* serovars, phage-1252 was selected for this study as the primary phage strain.

Furthermore, whole-genome sequencing of phage-1252 consists of double-stranded DNA, and its genome size is 244,421 bp with a G + C content of 48.51% (Table 3). Bioinformatics analysis revealed that phage-1252 belongs to the genus *Duplodnaviria* in the order Myoviridae family (NCBI data), and it showed the highest percentage (97.65%) of similarity to the previously known Salmonella phage SPN3US (NCBI:txid1090134), Enterobacteria phage SEGD1 (NCBI:txid1805456), and Proteus phage-7 (NCBI:txid2767546), respectively. The genome comparison of phage-1252 and previously known Salmonella phage SPN3US, Enterobacteria phage SEGD1, and Proteus phage-7 is shown in Figure 1a,b. As shown in the phylogenetic tree, phage-1252 and other similar known phages such as Salmonella phage SPN3US, Enterobacteria phage SEGD1, and Proteus phage-7 are separated into different branches of the phylogenetic tree (Figure 2) and showed different divergences. These results revealed that phage-1252 is a novel phage.

The phage-1252 genome was analyzed through PHASTER and RAST which showed that integrase genes were not identified in the phage genome. These results demonstrate that phage-1252 is likely lytic phages. In addition, according to our analysis using the database platform of VFPB and CARD, pathogenic genes or virulence properties as well as antibiotic resistance genes were not detected in the full-length genome of phage-1252. These findings suggest that phage-1252 may not carry risk in transferring genes to other microbes in a complex microbial ecosystem, and it is biologically safe for practical use.

### 3.3. Phenotyping Characteristics of the Phage-1252

After enrichment and purification, phage-1252 could lyse, and showed plaques on the bi-phasic agar plates. The plaque morphology of phage-1252 was clean and small but varied in size. The plaque diameters of phage-1252 are approximately 2.5 mm to 0.5 mm on the bi-phatic agar plate, with a lawn of BHC (Figure 3a). Phage-1252 has a strong lytic effect on BHC. We used MOI = 1 to determine in vitro lytic activity within different time periods affecting BHC. The data show that phage-1252 inhibited BHC growth after 6 h of treatment and slowed the growth of BHC approximately 25% after 24 h (Figure 3b). The growth curve for phage-1252 was found by incubating it with BHC in LB broth at a MOI of 0.1 to ensure that each phage could attach and infect a bacterium. Results showed that the latent and rise periods for the phage were approximately 40 and 30 min, respectively. The burst size was estimated to be 56 PFU/cell (Figure 3c). Testing of the thermal stability of the phage by inoculating in increasing temperatures showed phage-1252 to remain relatively stable and maintain original activity between 4 °C and 55 °C for 1 h. The averaged titer was 13 log_10_ PFU/mL upon exposure at different temperatures for 1 h, with no statistical differences between temperatures (Figure 3d). However, after incubating at 60 °C for 1 h, no plaque was detected in the plates, suggesting that phage-1252 cannot maintain its lytic activity when the temperature is 60 °C or over.

## 4. Discussion

Phage therapy was reported and considered a promising therapy for bacterial infection after phages were discovered in 1915, even though little was understood of their biological function or their strengths and limitations [33]. Due to the dramatic effectiveness of synthetic antibiotics, phage therapy had a relatively short history in the early 20th century. As antimicrobial agents, phages have been widely used in human and animal therapy, and in the food industry for food safety [15,34]. Nonetheless, the narrow host range is the main restriction when using phages as a biological control for *Salmonella*, which could have an advantage within certain applications but also has its drawbacks. On the one hand, narrow host-range phages conserve the native microbial flora and reduce the risk of community-wide resistance. On the other hand, such phages may need to be isolated from each specific type of bacteria, increasing the time and effort needed to isolate and identify them. Considering these facts, treating a bacterial infection with phage cocktails has been validated [35]. As a result, broad host-range phages could be a promising answer [36]. The range of phages has been linked to the attachment stage of phage infection, as many phages are extremely specific to the surface receptor of a particular bacterial host cell. According to recent studies, monovalent phages that bind to a single receptor are more likely to have a narrow host range than polyvalent phages [37]. Most isolated and published phages are usually specific to *S*. Typhimurium or *S*. Enteritidis, rather than recognizing multiple serovars or other minor *Salmonella* serovars [31].

Although genetic engineering could create broad host-range phages and enable them to maintain long-term suppression of bacterial growth in vitro [38], this approach would require advanced technology; therefore, isolating phages from nature remains an effective method for discovering and isolating novel species that may also have the ability to target a wide range and different serovars with high lytic capacity. To be useful for phage therapy, phages must be isolated from the environment and identified as having the ability to work against strains of the target bacterial pathogen, an obligately lytic ability, a broad host range, a lack of undesirable toxin genes, and the potential to generate a lysogen [36,39]. Generally, when looking for an effective phage from the environment, we have to consider where the host is. For this study, the target pathogen is *S. enterica*, an avian or a mammalian intestinal bacteria that can be easily isolated from animal feces, farm lagoons, and farm run-off water [36]. Therefore, in this study, a total of 20 *Salmonella* phages were isolated from animal drinking water, feces, and lagoon liquids collected from poultry and dairy farms in Maryland, USA, and their lyric activities were measured against various serovars of *S. enterica*. The phage-1252 was isolated from the lagoon water and showed highly effective lysing of six important serovars of non-typhoidal *S. enterica* (Typhimurium, Enteritidis, Newport, Heidelberg, Kentucky, and Gallinarum). In addition, the phage-1252 was also highly effective in lysing a Shiga toxin-producing enterohemorrhagic *E. coli*. Ultimately, this phage showed incredible diversity and adaptability for alternative therapeutic agents against multiple serovars of non-typhoidal *Salmonella* and *E. coli*.

To evaluate the potency or the ratio of phages to bacteria, MOI is frequently employed to calculate the amount of phages that should be applied during dosing. The results in this study demonstrate that phage-1252 at an MOI of 1 reduced the titer of S. Enteritidis approximately from 10^9^ to 10^7^ CFU/mL after 6 h treatment. During the study, we found that the effect of the higher titer of phage with MOI (MOI = 10, 100, and 1000) was not significantly different (not shown in the figure). It was considered that the higher the titer of phage inoculation, the more efficient the phage was in controlling bacteria; however, in this study, we observed there are optimal dynamic interactions between phages and hosts. Generally, a higher titer of phages would raise the attachment on the surface of hosts and infect more bacteria, then decrease the bacterium concentration in a short period. Another possibility is that the phage titer is saturated, the phage receptors are occupied, and the bacteria reduction will not increase when the MOI values rise. In addition, a higher titer of phages also means a higher potential of phages to apply to the lysogenic cycle. In the lysogenic phase, a bacterium carries a phage DNA that is integrated into the chromosome of a bacterium as a prophage and causes the proliferation of the prophage during replication and binary fission of bacteria, rather than lysis of the bacterium. Generally, the lysogenic bacterium is immune to subsequent infection by other phages through superinfection exclusion systems that prevent the same strain phages’ genome from entering into a host cell [40,41]. The possible explanation for bacterial growth increases over time could be the emergence of phage resistance mechanisms and phage-resistant mutant strains [40,42].

According to the analysis of a complete nucleotide sequence of phage-1252, its genome size is around 244,421 bp. It is similar to the published Salmonella phage SPN3US, a virulent phage effective against *S. enterica* and a few *E. coli* O157:H7 strains [43], Enterobacteria phage SEGD1, and Proteus phage 7. Phages containing a large genome size from 200 to 500 kbp are known as a “giant phages” or “jumbo phages”. Although phage-1252 is similar to these three published phages, most of its genome functions are still poorly understood [44,45]. In this study, we successfully analyzed a complete nucleotide sequence of phage-1252. Whole-genome sequencing and phylogenetic analyses supported phage-1252 as a novel phage strain. To gain more insight into the host-lysis mechanism of phage-1252, we screened its gene by NCBI-BLAST and compared it to the gene map of Salmonella phage SPN3US [46]. However, most phage genes have unknown functions, and their essential status requires further confirmation [46].

The major concern of phages as a therapy is the horizontal gene transfer between bacterial genomes through generalized and specialized gene transduction [47]. In generalized transduction, the phages can pick up any fragment of bacterial DNA and package it into the capsid of phages during assembly inside the host. They may then transfer it to new hosts after subsequent infection [47]. This is difficult to avoid in phage therapy because this gene transduction can also be carried out by lytic phages [39]. In contrast with specialized transduction, which frequently occurs with temperate phages, the phages pick up specific genes adjacent to the phage genome of the host’s chromosome. It can be excised together with the phage genome and integrated into the chromosome of new bacterial hosts [48]. For phage therapy, strictly lytic phages are generally preferred over temperate phages. Therefore, integrase genes, virulence factors, or toxin genes must be taken into consideration before the therapeutic use of phages, because some phages might be toxic to animals or humans. The lack of integrase, virulence, and antibiotic resistance genes being detected phage-1252 suggests that it is an appropriate candidate for phage therapy or food applications. 

## Figures and Tables

**Figure 1 foods-12-00797-f001:**
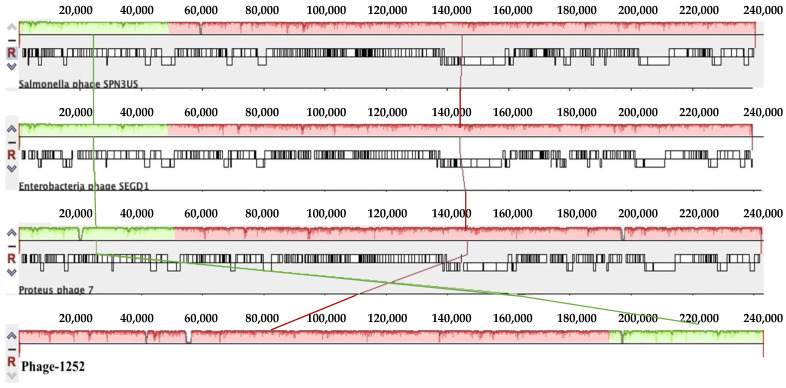
(**a**): Mauve alignment of the annotated complete genomes of phage-1252 with Proteus phage 7, Enterobacteria phage SEGD1, and Salmonella phage SPN3US (from bottom to top). Colored blocks correspond to similarity plots which indicate the degree of sequence similarity between distinct phages, while the height of the plot corresponds to the average nucleotide identity. Sections lacking homology are shown in white inside or outside the blocks. (**b**): Circular map of the phage-1252 genome using CGview. Outer ring (blue) corresponds to the phage-1252 genome. Subsequent rings compare Salmonella phage SPN3US (green), Enterobacteria phage SEGD1 (red), and Proteus phage 7 (purple). The innermost ring shows GC content and GC skew, in which outward peaks correspond to the positive and inward peaks correspond to the negative.

**Figure 2 foods-12-00797-f002:**
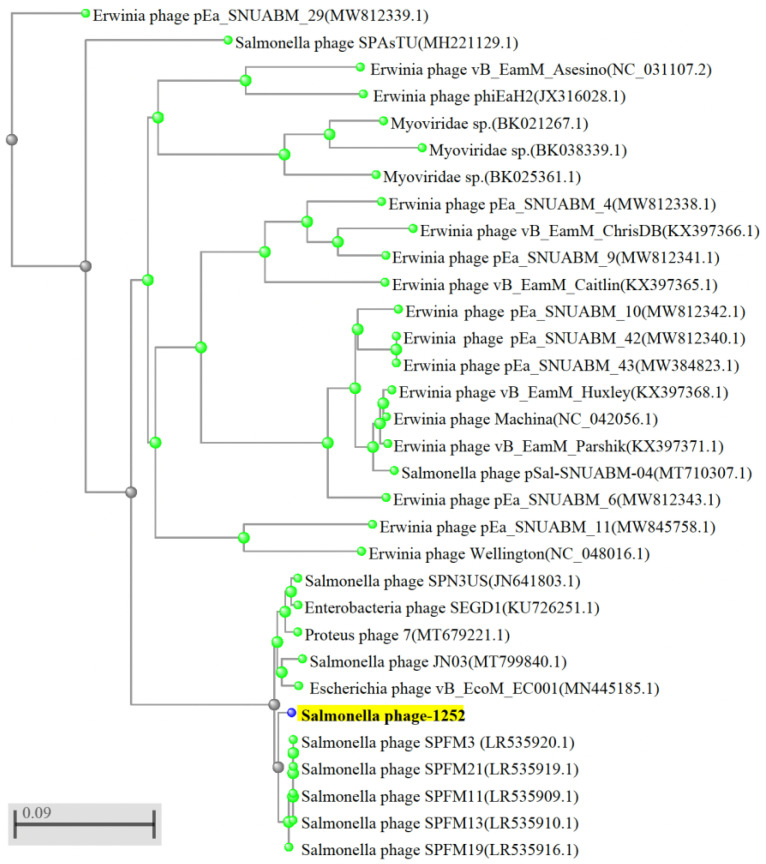
Phylogenetic analysis of *Salmonella* phage 1252. Phylogenetic tree generated by NCBI-BLAST using the complete genome sequence. *Salmonella* phage 1252, indicated with a red arrow, shows similarities in topology with other annotated *Salmonella* phages.

**Figure 3 foods-12-00797-f003:**
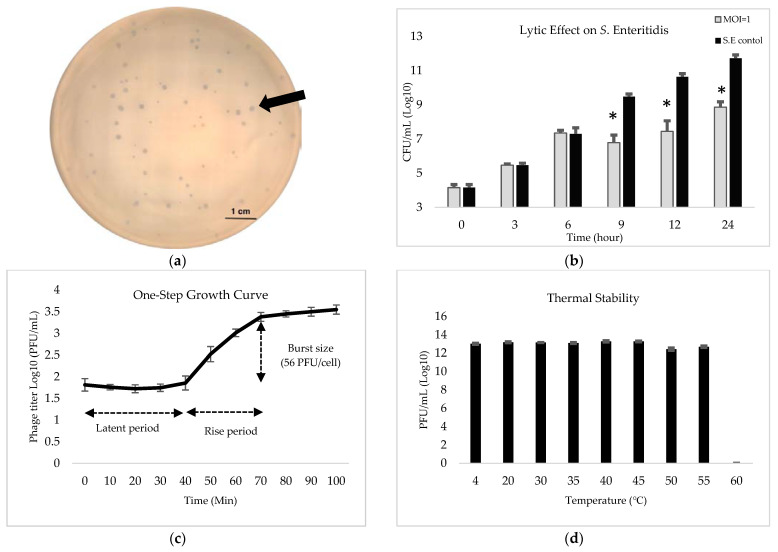
(**a**): Phenotypic and biological characteristics of phage 1252. Phage plaque formed on bi-phasic agar plates when cultured in a lawn of bacterial host cell (*S.* Enteritidis). Arrows indicate plaques of different sizes (2.5mm to 0.5mm). (**b**): In vitro lytic activity of phage-1252 in reducing the growth of BHC. The growth of untreated BHC (black) was compared to that inoculated with MOI = 1 of phage-1252 (grey). Error bars indicate the standard deviation from three trials. Significance is indicated by * (*p* < 0.05). (**c**): One-step growth curve for measuring titers of phage-1252 across time with BHC as the host when incubated in LB medium at MOI of 0.1. Error bars indicate the standard deviation from three trials. (**d**): Temperature stability of phage-1252 when incubated at increasing temperatures for 1 h.

**Table 1 foods-12-00797-t001:** Bacterial genera and species tested for phage in this study.

Genera/Species	Strain ID	Source
*S.* Typhimurium	LT2	ATCC
*S.* Enteritidis	13076	ATCC
*S.* Newport	-	Farm isolation
*S.* Kentucky	-	Farm isolation
*S.* Heidelberg	-	Farm isolation
*S.* Gallinarum	-	Farm isolation
*E. coli* O157:H7	EDL933	ATCC

**Table 2 foods-12-00797-t002:** Lytic activity of isolated phages against selected bacterial hosts.

Strains	Phage-1252	Phage-1336	Phage-2902	Phage-CF	Phage-CW
*S.* Typhimurium	+++	+++	++++	+	+++
*S.* Enteritidis	++	+++	++++	+++	++
*S.* Newport	++	++	++++	+++	++
*S.* Kentucky	++	++	+++	++	++
*S.* Heidelberg	+	+	++	++	++
*S.* Gallinarum	++	++	++++	+++	+++
*E. coli*	++	++	++	+++	+++

“++++”, complete lysis; “+++”, clearing throughout, but with slight turbidity; “++”, heavy turbidity with clear ring; “+”, heavy turbidity w/o ring or pinpoint plaque.

**Table 3 foods-12-00797-t003:** Whole-genome sequencing was analyzed by the Average Nucleotide Identity (ANI) calculator and NCBI-BLAST.

	Genome Size (bp)	A	C	G	T	GC Content (%)	Number of ORF	Genomic Similarity (%) (BLAST-Percent Identity)				
								Phage-1252	Phage-1336	Phage-2902	Phage-CF	Phage-CW
Phage-1252	244,421	64,036	57,739	60,821	61,825	48.51	432	-	100	100	98.13	98.15
Phage-1336	243,594	63,796	57,590	60,665	61,543	48.55	431	100	-	100	97.92	98.04
Phage-2902	296,802	75,834	72,833	69,924	78,211	48.1	546	100	100	-	97.98	97.94
Phage-CF	295,964	77,999	69,711	72,525	75,729	48.06	530	98.13	97.92	97.98	-	100
Phage-CW	250,077	63,478	61,681	59,825	65,093	48.59	428	98.15	98.04	97.94	100	-

## Data Availability

Data supporting reported results are available upon request, and genome sequencing and assembly of the phage-1252 can be found on NCBI Bioprojects (BioProject ID: PRJNA906125).

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
