# Peer review of "A Highly Effective Bacteriophage-1252 to Control Multiple Serovars of Salmonella enterica"

_foods, 2023, doi:10.3390/foods12040797_

Round 1
Reviewer 1 Report
Bearing in the mind antimicrobial resistance as huge public health issue development of potential alternatives to conventional antimicrobial treatment represents one of the major attempts in order to combat this problem. One of the method that gained more and more attention worldwide is phage therapy making this manuscript valuable and interesting. According to recent reports, several serovars of Salmonella, specifically Typhimurium and Enteritidis, which are also a severe threat to public health, have been aggravated by their increasing antibiotic resistance pattern. Phage therapy was reported and considered a promising therapy for the bacteria infection after phages were discovered. The biological function or its strengths and limitations are explained in this manuscript. Moreover, in the manuscript is highlighted the major concern of phages as a therapy, is its horizontal gene transfer. It will be valuable to add in conclusion further activities regarding the implementation of the phage therapy.
Author Response
"Please see the attachment."

Reviewer 2 Report
The author isolated several phages with a wide host range, which exhibited lysis ability on multiple serovars of Salmonella and Escherichia coli. They also made a simple genome analysis and the characteristics of phages. However, there confusing results, many mistakes in language, and lack of innovation and important data, led me to the decision that this article needs a major revision.
Major comments:
Too much background in abstract, and the phenotypic characteristics of Phage-1252 should be included.
Line 13-14: this sentence should be revised, and resistance should be resistant
There is no mention to search of any "integrase" genes, for excluding lysogenicity. This analysis must be done.
table 2 should be revised and included in the supplementary material. or removed?
Line 215-216 four phages are identical? are they isolated from the same source? or contamination among samples.? the author should provide the source information of these purified phages and make an explanation on it.
five identical phages should be changed to just one phage! Besides spot tests, EOP analysis of phages should be included among these different serovars.
Importantly, for example, why do host range differ between phage1216 and phage 1336 despite their identical genome? so confusing!! contamination in phages?
Introduction should be revised and Discussion should be rewritten, too much background and unrelated information.
Genus name should be italicized, e.g., Salmonella, Proteus, Enterobacteria
Full names and abbreviations should be revised throughout the article.
Other minor suggestions were included in PDF version.

Author Response
"Please see the attachment."

Reviewer 3 Report
In this manuscript, Tung et al isolated a bacteriophage strain that is able to lyse multiple serovars of the food borne bacteria Salmonella enterica. They perform extensive phenotyping of this novel bacteriophage as well as carry out whole genome sequencing to understand its origin. Phylogenetic analyses revealed that it is indeed a novel phage strain. Sequencing also revealed that this phage does not contain antimicrobial resistance genes, virulence or toxin factors, hence this phage is a promising candidate for phage therapy or food applications. The work is important as Salmonella enterica infections pose a significant challenge for the health care, and even in the context of phage therapy, most phages are very species specific. Moreover, development of antibiotic resistance bacteria is a big problem.
There are a few minor points that need to be addressed:
1. From Table 3, it seemed to me on face value that phage 2902 was the most interesting one, however I could not find an explanation as to why 1252 was chosen for subsequent analyses. This is clearly explained in the Discussion, however I think this section needs to be moved to the Results.
2. Table 4: The phage similarity column has only three entries and they are not aligned. This is quite unclear.
3. Last paragraph of the discussion: the distinction between generalized and specialized transduction is helpful. Do we know what type of transduction happens with the phages isolated in this study?
4. General question: phage 2902 which happened to be a mix of phages apparently lyse multiple serovars of Salmonella very well. Why is then such a cocktail not useful for therapy? Are there any pitfalls one should be aware of?
Author Response
"Please see the attachment."

Round 2
Reviewer 2 Report
This manuscript still has many errors and important and necessary data missing. I also suggest the authors check the whole manuscript thoroughly.
All five phage genome sequences analyzed in this manuscript should be submitted to public database eg. NCBI. The author only uploaded the data for phage 1252, but they sequenced 8 phages and made a simple analysis in this study.
Results 3.1, line 205-211, I am still confused on phage isolation or phage-positive samples in this paragraph. A total of 20 phages were isolated?? are they just 20 Salmonella-phage positive samples using S. Enteritidis as a bacterial host through spot test? The author selected 8 samples to purify based on the lysis profile of phage-positive samples? Or 20 phages were isolated and purified, but 8 phages were selected for broad host range?? this paragraph should be revised.
Line 219, the authors mentioned that genome sequence data of 8 phages were shown in Figure1. However, figure 1 just showed the comparison of phage 1252 and other known phages. The title of table 3 should also be revised.
Line 226-230, why Salmonella phage SPN3US, Enterobacteria phage SEGD1, and Proteus phage were selected for comparison? I suggest to revise these sentences, maybe the highest identity to phage X (percent %) via Blast analysis.
Table 3 just showed the genome size and GC content, at least Genbank accession number , annotated ORFs with known and unknown predicted functions should also be added.
Line 232-243, the author thought that phage 2902 was a combination of their phage-1252 and known phage PEA2-3, the similarity percent between them showed be provided separately. 100% identity to phage 1252? how about phage PEA2-3? what about the core genes encoding proteins involved in DNA replication, DNA packaging, morphogenesis, and lysis? the core genes with same function from two phages were all present in phage 2902? And the english language in this paragraph should also be rewritten.
line 222-223, the isolation source of three identical phages should be added to make it more clear.
Line 224, remove “for”?
Figure 2, The phage names on the last 8 branches need to be modified, the last phage miss phage name.
One thing I should point out that a virus name should not be italicized, even when it includes the name of a host species or genus, and should be written in lower case. The rules for naming virus taxa can be found in the ICTV Code<http://ictv.global/code>. I am sorry for the suggestion before.
Other minor revisions can be found in the attached pdf.

Author Response
"Please see the attachment."
